# Associations between Second-Hand Tobacco Smoke Exposure and Cardiorespiratory Fitness, Physical Activity, and Respiratory Health in Children

**DOI:** 10.3390/ijerph182111445

**Published:** 2021-10-30

**Authors:** Melissa Parnell, Lawrence Foweather, Greg Whyte, John Dickinson, Ivan Gee

**Affiliations:** 1Public Health Institute, Liverpool John Moores University, Liverpool L2 2QP, UK; melissajparnell@hotmail.com (M.P.); I.L.Gee@ljmu.ac.uk (I.G.); 2Research Institute for Sport and Exercise Sciences, Liverpool John Moores University, Liverpool L3 3AF, UK; G.Whyte@ljmu.ac.uk; 3School of Sport and Exercise Sciences, University of Kent, Medway ME4 4AG, UK; j.w.dickinson@kent.ac.uk

**Keywords:** cardiorespiratory fitness, physical activity, second-hand smoke, children, VO_2peak_, respiratory health, low socioeconomic status

## Abstract

Background: Cardiorespiratory fitness (CRF) and physical activity (PA) are associated with a plethora of positive health effects. Many UK children fail to meet the recommended level of PA, with an observed decline in CRF levels over recent decades. Second-hand tobacco smoke (SHS) is responsible for a significant proportion of the worldwide burden of disease, but little is understood regarding the impact of SHS exposure on CRF and PA in children. The aim of this study was to test the associations between SHS exposure and CRF, PA, and respiratory health in children. Method: Children (9–11 years) from UK primary schools in deprived areas participated (n = 104, 38 smoking households). Surveys determined household smoking, and exhaled carbon monoxide was used to indicate children’s recent SHS exposure. CRF (VO_2peak_) was assessed via maximal treadmill protocol using breath-by-breath analysis. Fractional exhaled nitric oxide and spirometry were utilised as indicators of respiratory health. Results: Linear regression models demonstrated that SHS exposure was negatively associated with allometrically scaled VO_2peak_ (*B* = −3.8, *p* = 0.030) but not PA or respiratory health. Conclusion: The results indicate that SHS is detrimental to children’s CRF; given that approximately one-third of children are regularly exposed to SHS, this important finding has implications for both public health and the sport and exercise sciences.

## 1. Introduction

### 1.1. Cardiorespiratory Fitness and Physical Activity

Cardiorespiratory fitness (CRF) is a health-related component of physical fitness and defined as the ability of the circulatory, respiratory, and muscular systems to supply oxygen during sustained physical activity [1]. Accordingly, CRF is commonly employed as a global measure of health [1,2]. CRF is an important marker of physical and mental health, as well as academic achievement in youth [3,4], which reinforces the importance of early intervention efforts to promote CRF. In high and upper-middle income countries, there was a substantial decline in CRF for children and adolescents between the 1980s and 2000 [5]. In the North West of England, CRF in children has been decreasing since the 1990s [6,7]. Low levels of CRF and a temporal reduction in CRF are therefore suggestive of a decline in population health [5]. Physical activity (PA), in particular moderate-to-vigorous-intensity PA (MVPA), is positively associated with CRF [8,9], and low PA in childhood is predictive of low PA in adulthood [10,11]. The United Kingdom (UK) Chief Medical Officers’ guidelines state that children and youth aged 5–18 years should achieve at least an average of 60 min of MVPA daily [12], yet less than half of all children and young people met these guidelines in England in 2019 [13]. Lower socioeconomic status (SES) has been shown to be associated with lower levels of PA [14,15] and CRF [16,17] in youth. It is therefore important to understand factors that influence CRF and PA in youth from areas of high deprivation to design effective interventions.

### 1.2. Second-Hand Tobacco Smoke

Tobacco smoke is a toxic and carcinogenic mixture of over 5000 different chemicals [18]. Second-hand smoke (SHS), often referred to as environmental tobacco smoke, is composed primarily of smoke that emanates from the end of the burning cigarette (sidestream smoke), smoke that the smoker inhales and exhales (mainstream smoke), and contaminants that diffuse through the cigarette paper [19,20]. SHS is responsible for a substantial proportion of global mortality and morbidity for both adults and children [21,22], with 1.2 million deaths attributable to SHS exposure in 2017, of which 5% occurred in children under 10 years [23]. Children are particularly susceptible to the effects of SHS due to their high respiratory rates and immature organs [24]. Exposed children are at increased risk of chronic airway inflammation, lung function defects [25], severe asthma attacks, respiratory infections, ear infections, sudden infant death syndrome [26], and increased risk of hospitalisation in asthmatics [27].

The Smoke-free Legislation, which came into effect in England in July 2007 as a result of the Health Act 2006 [28], made it illegal to smoke tobacco in enclosed places, though smoking is still permitted in private residences and open public places. Indeed, two main determinants of children’s SHS exposure are smoking by parents or caregivers and whether smoking occurs in the home [29]. In 2019, 15.9% of UK men (3.8 million) and 12.5% of women (3.1 million) reported being current smokers [30]. While children’s exposure to SHS has declined in England by 79% since 1998 due to the emerging social norm of smoke-free homes [31], UK-based studies have shown approximately 31.5% of children to have detectable levels of salivary cotinine, an indication of recent tobacco smoke exposure [31], with 96.9% of children from the poorest families demonstrating detectable levels of salivary cotinine [32]. Children from low socioeconomic status (SES) are therefore more likely to be exposed to SHS [32,33], and consequently more likely to suffer the detrimental impacts of SHS exposure.

### 1.3. Second-Hand Tobacco Smoke, Cardiorespiratory Fitness, and Physical Activity

The impact of SHS exposure on cardiorespiratory fitness has been studied in adults, with SHS exposure associated with reduced exercise performance [34,35] and reduced VO_2max_ [36]. Two main constituents of cigarette smoke, nicotine and carbon monoxide (amongst others), exhibit toxic effects on cardiovascular function both at rest and during exercise in adults [37]. For children and young people, research is limited. Of the research that exists, studies have found children’s CRF (measured by maximal cycle ergometer test) to be significantly reduced for those with smoking parents [38], adolescents exposed to SHS to have increased systolic blood pressure whilst exercising [39], and obese children exposed to SHS were found to have reduced performance on a six-minute walk test [40]. PA has been shown to reduce adolescent smoking uptake [41] and aid cessation [42], but there is limited research exploring the association between SHS exposure, or having a smoking family member, on children’s engagement in PA. Further research is therefore required to examine the association between SHS exposure, cardiorespiratory fitness, and PA in children. Given that the respiratory system is important for CRF and PA, and that SHS exposure has been found to be associated with increased risk of respiratory disease [25,26,27], examining the association between smoking exposure and markers of respiratory health is also important. Understanding whether household SHS exposure is detrimental to children’s CRF, PA, and respiratory health will be of value in the domain of public health and could provide novel pathways for health promotion interventions.

### 1.4. Aim 

The aim of this study was to examine whether SHS exposure in the home is associated with children’s (i) cardiorespiratory fitness, (ii) participation in physical activity, and (iii) markers of respiratory health.

## 2. Materials and Methods

### 2.1. Study Design

This cross-sectional study was granted ethical approval by the University Research Ethics Committee (Ref: 16/PBH/001) and comprises the quantitative aspect of a larger mixed-methods research project [43]. Data collection began in September 2017 and ended in February 2019, with schools participating at different timepoints throughout the year and determined according to convenience relative to the schools. The wider project recruitment and data collection process is demonstrated in the Appendix A.

### 2.2. Participants and Setting

Participants were targeted as being aged 9–11 years old and in year 5 or 6 at a state-funded primary school within two metropolitan boroughs in Northwest England. This age group was targeted since evidence from Northwest England has reported low fitness among primary school children [6], and therefore this study sought to examine whether smoking exposure was a factor. One-hundred-and-forty-seven primary schools were contacted across the region via an email containing study information, followed by a phone call. Gatekeepers (headteachers) were provided with information sheets and face-to-face meetings were organised with representatives from primary schools. Participating schools received informational presentations, during which children were free to ask questions about the study. Information packs containing participant information sheets for children and parents, child medical questionnaires, parental surveys, and parental consent and child assent forms were then distributed to all parents and guardians. Parental consent, child assent, and completed medical questionnaires were required to participate in the study. Exclusion criteria were any medical conditions that limit a child’s ability run on a treadmill.

### 2.3. Data Collection Procedures

Following parental consent, child assent, and completed parental surveys, exhaled carbon monoxide measurements were taken at the primary school, on the morning of the visit to the University laboratories. Participants were then transported to the laboratories in small groups of 3–4 children during school time (09:00–15:00). Once at the University, participants’ anthropometrics were measured. Respiratory health markers including fractional exhaled nitric oxide measurements and spirometry were subsequently taken. Finally, children participated in cardiorespiratory fitness testing once all respiratory measures had been completed. In between laboratory measures, children completed a self-report physical activity survey.

### 2.4. Measures

#### 2.4.1. Demographic Information

Participant (adult and child) demographic information, including parental education, child age and ethnicity, and home postcode, was obtained via the parental survey. Household deprivation was assessed via the English Indices of Multiple Deprivation (EIMD) [44] using the participant’s home postcode and the Ministry of Housing, Communities, and Local Government postcode look-up tool [44].

#### 2.4.2. Second-Hand Smoke Exposure

Parents or guardians self-reported household smoking status in the parental survey. Items were adapted from the well Global Adult Tobacco Survey (GATS) by the Global Adult Tobacco Survey Collaborative Group [45] and determined the number of tobacco smokers living in the home, as well as the rooms in which smoking occurred and/or was permitted, and how many cigarettes were smoked each day per person. The GATS is a well validated and reliable, globally standardised survey used to collect tobacco-related information [45]. Space was provided for participants to include information regarding smoking habits for up to four members of the household, with more space available upon request. Similar information was collected for e-cigarette use. Participants were classified into ‘non-smoking household’ or ‘smoking household’ according to whether a household member smoked cigarettes or not, regardless of where smoking was permitted. Households with an adult that used e-cigarettes were classified as non-smoking, as no significant differences in fitness and health outcomes were observed between children from non-smoking and e-cigarette-using households. Households were then further classified as permitting smoking ‘indoors’ or ‘outdoors only’. 

Exhaled carbon monoxide (eCO) was measured in ppm using a breath Smokerlyzer PiCO device (Bedfont, UK) to determine children’s recent tobacco smoke exposure. Previous studies have shown eCO to be a useful indicator for recent tobacco smoke exposure in adults [46,47,48] and youth [49]. A threshold of 3–4 ppm is suggested to distinguish smokers from non-smokers [46], but there is currently little research regarding thresholds for children who are exposed to SHS. Pilot work indicated that eCO declined within several hours post-tobacco smoke exposure. Therefore, carbon monoxide (CO) measurements were taken on the morning (08:30) of laboratory visits, at school prior to departure, to better reflect second-hand smoke exposure from the home (mean value of two attempts). Participants were asked to hold their breath for 15 s before exhaling continuously with a constant force into the Smokerlyzer mouthpiece.

#### 2.4.3. Anthropometry

All anthropometric assessments were conducted on arrival to the University laboratories in accordance with the standards of the International Society for the Advancement of Kinathropometry [50]. Children were assessed whilst wearing light clothing and shoes removed. Body mass to the nearest 0.1 kg (Seca, Birmingham, UK), stature, and sitting stature to the nearest 0.1 cm (Seca, Birmingham, UK) were assessed using standard techniques [51]. Body mass index (BMI) was calculated from stature and mass (kg·m^2^) with age- and sex-specific International Obesity Task Force BMI cut-offs used to classify child BMI and weight status for descriptive purposes [52]. Years to peak height velocity, a somatic indicator of physical maturity, was estimated using stature, sitting height, and body mass [53]. 

#### 2.4.4. Physical Activity

Children completed the Physical Activity Questionnaire for Children (PAQ-C) [54]. The PAQ-C requests responses for the last 7 days by asking participants to check a list of activities for frequency of participation, including PA during school time and out of school [55]. Each question is scored from 1 to 5, for example, Question 10: On the last weekend, how many times did you participate in sports, dance, or play games in which you were very active? (Tick one). ‘None’ would equal a score of 1, and ‘6 or 7 times last week’ would give a score of 5. A mean is calculated for all questions to give an overall PAQ-C score. The PAQ-C is a validated and reliable measure of PA levels in children regularly used in PA surveillance [54,55,56].

#### 2.4.5. Respiratory Health 

Fractional exhaled nitric oxide (FeNO), an indication of airway inflammation, was measured (mean value of two attempts) at rest using a NIOX^®^ VERO device (Circassia, UK). This measured FeNO in the exhaled breath, at rest, at a constant flow rate of 50 mL/min. FeNO was performed prior to spirometry measures to avoid potential carryover effects [57] and taken as the mean of duplicate measures [58]. FeNO offers additional benefits to spirometry by detecting eosinophilic airway inflammation [58], an indication of asthma [59]. The non-invasiveness and instantaneous result make FeNO a suitable method for assessing lung health in children [60]. For children, the following FeNO thresholds were used: <20 ppb (low), 20–35 ppb (intermediate), and >35 ppb (high) [58].

Spirometry measures including forced vital capacity (FVC), forced expiratory volume (FEV1), peak expiratory flow (PEF), and forced expiratory ratio (FER) were taken at rest using a digital micro-spirometer (Micro-plus spirometer, CareFusion, UK). Measurements were made in triplicate and the best value compared against predicted values for sex, height, and age [61]. Spirometry values were normalised by a factor of 0.9 for black children and 0.95 for children of other ethnicities [62]. Spirometry values were then expressed as percentages of predicted for sex, age, height, and ethnicity. An FEV1 < 80% and FER < 70% predicted was considered obstructive, and FEV1 < 60% and FVC < 60% was considered restrictive [63]. Information regarding respiratory disease including asthma (and general medical background) was collected via medical questionnaire.

#### 2.4.6. Cardiorespiratory Fitness

Cardiorespiratory fitness (CRF), using peak oxygen uptake (VO_2peak_) as a marker, was assessed using an individually calibrated, discontinuous incremental treadmill test to volitional exhaustion using breath-by-breath analysis (Jaeger Oxycon Pro, Viasys Health Care, UK). Peak oxygen uptake (VO_2peak_), as opposed to VO_2max_ (maximal oxygen uptake), was used as children often fail to reach a plateau [64]. Previous studies have demonstrated that ‘true’ VO_2max_ values can be achieved in children without the need for plateau as long as test endpoints are met [65]. The following method is based on an established protocol and is supported by previous work [66].

A paediatric facemask (Hans Rudolph, Kansas City) covering the nose and mouth was secured via an adjustable nylon harness prior to the test beginning. Before using the treadmill, participants wore a specialised harness that would cause the treadmill to stop in case of any trips or falls. Children underwent a familiarisation period of walking and running on the treadmill prior to the test. To account for differences in age and limb length, VO_2peak_ test speeds were individually calibrated by anchoring treadmill speeds to set Froude (Fr) numbers [67]. Participants completed 2-min stages, stage one speed at Fr 0.25, stage two speed at Fr 0.5, with each additional stage determined by the difference in speed for stages one and two (~2 km/h). The treadmill remained at 1% gradient throughout. The test was terminated at the point of volitional exhaustion when the participant was unable to continue despite strong verbal encouragement.

Oxygen uptake (VO_2_) and carbon dioxide production (VCO_2_) were measured breath-by-breath with an online system (Jaeger Oxycon Pro, Viasys Health Care, Warwick, UK). Prior to each testing session, the Oxycon Pro system was calibrated using known volumes of gases (0.5% CO_2_ and 20.5% O_2_) and known volumes (3.0 L/s). Heart rate was monitored continuously (Polar, Kempele, Finland). 

VO_2peak_ was defined as the highest 15 s averaged oxygen uptake achieved during the test when participants reach volitional exhaustion, and the below endpoints met. VO_2peak_ was accepted as the maximal index when participants exhibited any of the following subjective indicators of maximal effort; unsteady gait, hyperpnea, facial flushing, sweating, in addition to objective indicators: respiratory exchange ratio (RER) > 1.05 and heart rate > 199 beats/min [66].

The Pictorial Children’s Effort Rating Table (PCERT) [68] was used to establish participants’ perceived exertion. The PCERT uses pictures as well as numbers and descriptive language, reflecting the changing physiological demands of the exercise task in a child-friendly format [69]. Participants were asked to state or point to the point on the scale that best described their effort rating at the end of each two-minute stage.

Most research to date has expressed CRF as VO_2peak_ ratio scaled for mass (mL·kg^−1^·min^−1^), but expressing CRF in this way over-scales for mass and leads to spurious correlations with other health-related outcomes [70]. CRF was presented as allometrically scaled VO_2peak_ using a sample-specific calculated mass exponent (mL·kg^−0.53^·min^−^^1^). Data for absolute VO_2peak_ (mL·min^−1^) and VO_2peak_ ratio scaled by body mass (mL·kg^−1^·min^−1^) are provided within the Appendix A. Mass exponents were calculated using log-linear regression models of mass and absolute VO_2peak_, as described by Welsman and Armstrong [70], where the generated ‘b’ is the mass exponent. Sex- and age-group-specific exponents were not calculated due to the small sample size. The generated mass exponent (0.53) was ‘tested’ by correlational analysis between allometrically scaled absolute VO_2peak_ and mass, which was found to be close to zero (r = −0.046, *p* = 0.663), indicating the influence of mass was successfully removed.

For descriptive purposes, participants were classified as fit or unfit according to published thresholds for identifying aerobic fitness and associated cardio-metabolic disease. Thresholds used were taken from the review by Lang et al. [71], which incorporates data from 1,142,026 youth from 50 countries. VO_2peak_ scores below 42 mL·kg^−1^·min^−1^ for boys and 35 mL·kg^−1^·min^−1^ for girls indicate higher risk of cardiovascular disease and were classified as unfit.

### 2.5. Statistical Methods

Statistical analyses were performed using SPSS for Windows (version 26; SPSS, Chicago, IL, USA). Data are expressed as mean ± standard deviation unless otherwise stated. Data that were not normally distributed were base-10 logarithm transformed (EIMD), natural log transformed (FeNO), or square root transformed (total number of cigarettes smoked per day) prior to analysis, although are presented pre-transformation, with geometric mean and geometric standard deviation, in descriptive tables for ease of interpretation. Differences by sex and household smoking status were assessed using independent sample Student’s *t*-tests and are provided within the Appendix A.

Unadjusted and adjusted multiple linear regression models were performed for allometrically scaled VO_2peak_, PA, and each of the markers of respiratory health (FEV1%, FVC%, PEF%, FER, and FeNO), to assess if these outcome variables were predicted by household smoking. The forced entry (enter) method was selected over the stepwise method in order to include known correlates based on theoretical knowledge and past research, and to ensure the researcher had control over what variables were entered into the models [72].

While several indicators of household smoking status and level of smoking were available, the number of cigarettes smoked per day per household was selected as a more precise measure of household exposure than the binary measure of household smoking status (smoking or non-smoking). Household smoking status was not entered into the regression models, in addition to the number of cigarettes smoked, due to the high correlation (r > 0.9) between these two variables. Exhaled CO was not significantly different for children from smoking and non-smoking homes (*p* = 0.215), and was not correlated with the number of cigarettes smoked per day (r = 0.157, *p* = 0.119); similar findings have been observed in children in previous studies [73]. Exhaled CO was therefore deemed inappropriate as a measure of second-hand smoke exposure in children for the present study and was not used in the predictive models. ANOVA and Tukey post hoc analysis determined that CRF, PA, and respiratory variables for children from non-smoking and e-cigarette-using families were not statistically different and therefore e-cigarette use was further classified as non-smoking.

All unadjusted models included the sole predictor of the square root transformed number of cigarettes smoked per day. Other variables were selected based on known determinants from previous research. For allometrically scaled VO_2peak_, sex, age, stature, maturation, PA, and deprivation (logEIMD) [74] were included in the adjusted model. Mass was not included in the models as mass is already accounted for within the calculation for allometric scaling. PA models contained known determinants of PA including sex, age, BMI, maturation, and deprivation (logEIMD) [75,76,77]. As spirometry measures, FEV_1_%, FVC%, and PEF% values were already adjusted for known determinants of lung function prior to modelling (i.e., sex, age, height, and ethnicity), adjusted linear regressions for these measures included mass, diagnosed asthma, and deprivation (logEIMD) [58,61,78]. Linear regressions for FER additionally included age, sex, and stature as FER values are not presented as percentages of predicted. Linear regressions for FeNO included sex, age, mass, stature, asthma diagnoses, and deprivation (logEIMD) [58,78].

For all models, there was linearity, as assessed by partial regression plots and a plot of studentized residuals against the predicted values. There was homoscedasticity, as assessed by visual inspection of a plot of studentized residuals versus unstandardized predicted values. There was no evidence of multicollinearity as assessed by tolerance values greater than 0.1. Studentized deleted residuals greater than ±3 standard deviations were excluded from absolute VO_2peak_ (n = 4), PEF% (n = 1), and FER (n = 2). There were no leverage values greater than 0.2 and no values for Cook’s distance above 1. The assumption of normality was met, as assessed by a histogram and P-P Plot.

## 3. Results

### 3.1. Participant Descriptives

#### 3.1.1. Sampling

Of the 147 schools contacted across the Merseyside region, four schools (two each from both Liverpool and Wirral areas) participated in the study (3% response rate). Schools that declined to participate provided a variety of reasons such as ‘too busy’, ‘no staff available to coordinate the project’, and ‘the project is too contentious due to the smoking focus’. Total participation (consent rate) was 26.7% with 104 children taking part (46 boys, 58 girls) out of a possible 390 invited from the participating schools. 

Out of the 104 participants with written parental consent and participant assent, ten children were excluded from the VO_2peak_ analysis for failing to reach ‘peak’ threshold criteria (n = 7), being unable to run on the day (n = 2), and one participant requested not to undertake the fitness assessment. In total, 94 children (43 boys, 51 girls) were included in the VO_2peak_ analysis. One participant requested not to be weighed or have their height measured. Four participants had no corresponding exhaled carbon monoxide data due to unavailability at the time of testing. Eleven participants failed to provide a home postcode, or the provided postcode did not generate an EIMD score, and therefore school postcode was used as a substitute. Two children failed to perform a successful FeNO test. In total, complete data were available for 92 participants. Participant characteristics are presented in Table 1. Descriptive data by gender and household smoking status are provided in the Appendix A.

#### 3.1.2. Participant Demographics and Weight Classification

The geometric mean English Indices for Multiple Deprivation (EIMD) rank was 1427 (geometric SD 5652) and most participants’ postcodes were within the first (69.2%) and second (16.3%) most deprived deciles. All four primary postcodes were within areas of very high deprivation (1st decile). The percentage of parents or guardians with no formal education was 3.3%, 33.7% were educated to high school level, 41.3% had completed college or sixth form, 13.0% had a Bachelor’s degree, and 8.7% had a Master’s degree or above. White British children made up 76.9% of the sample population, with 6.7% Black British, 2.9% White Polish, 1.9% White Portuguese, 1.9% Black African, 1% Black other, 1% Chinese British, and 7.7% other. Overall, out of 103 children (n = 58 girls and 45 boys), 35.0% were overweight or obese, including 28.9% of boys and 39.7% of girls. Further descriptive data regarding socioeconomic and weight status are provided in the Appendix A, respectively.

#### 3.1.3. Cardiorespiratory Fitness

Participants could be classified as fit or unfit according to established thresholds [71] based on ratio scaled VO_2peak_ (mL·kg^−1^·min^−1^). Using the CRF thresholds of 42 mL·kg^−1^·min^−1^ for boys and 35 mL·kg^−1^·min^−1^ for girls, 83.0% of participants were classified as fit, including 86.3% of girls and 79.1% of boys, which was not statistically different, Chi-square (1) 2.3, *p* = 0.354. Boys were found to have significantly higher allometrically scaled VO_2peak_ (t (92) = 3.6, *p* = 0.001) than girls. Further descriptive data regarding CRF and SHS exposure are provided in the Appendix A.

#### 3.1.4. Physical Activity 

The mean level of self-reported physical activity (PA) is presented in Table 1 and the Appendix A. No significant differences were observed between boys and girls for PA score (t (101) = 1.7, *p* = 0.099). Using a threshold score of 2.73 to classify children as active, 87.4% of children in the sample were classified as physically active. When split by sex, 95.7% of boys and 80.7% of girls were classified as physically active, which was statistically significant (Chi-square (1) = 5.2, *p* = 0.023). Further descriptive data regarding PA and SHS exposure are provided in the Appendix A.

#### 3.1.5. Spirometry

The mean spirometry (%) values for the sample are shown in Table 1. For all four spirometry measures, mean values were below the predicted values for sex, age, height, and ethnicity (equivocal to 100%) by 10.3–24.7%, indicating lower than predicted spirometry across the sample. No significant differences were observed between the mean spirometry values for boys and girls (Appendix A). Further descriptive data regarding spirometry and SHS exposure are provided in the Appendix A.

#### 3.1.6. Fractional Exhaled Nitric Oxide

The range for FeNO measurements was high with a minimum value of <5 ppb (below the detection limit of 5 ppb) and a maximum of 147 ppb. The mean concentrations of FeNO are presented in Table 1 and the Appendix A and were not statistically different between boys and girls (t (100) = 0.9, *p* = 0.384). FeNO levels could be classified as low, intermediate, and high according to established thresholds [58]. Most children (70.6%) had low levels (<20 ppb) of FeNO, 13.7% had intermediate levels (20–35 ppb), and 15.7% had high levels (>35 ppb). FeNO concentrations were not significantly different between children with diagnosed asthma (24.4 ± 12.7 ppb, n = 9) and those without asthma (20.9 ± 23.7 ppb), (t (96) = −0.6, *p* = 0.567). Further descriptive data regarding spirometry and SHS exposure are provided in the Appendix A.

#### 3.1.7. Household Smoking Status

Tobacco smoking only, by one or more members of the household, was reported in 35 households (33.7%). In addition, three parents reported using e-cigarettes in addition to smoking tobacco (2.9%), and parents from ten households reported using e-cigarettes only (9.6%). Therefore, a total of 38 (36.6%) households reported smoking tobacco. Neither smoking tobacco cigarettes nor using e-cigarettes was reported in 56 (53.8%) households. Of the 38 participating households that reported tobacco smoking, ten (26.3%) reported two people living in the home that smoked, with the remaining 28 households (73.7%) reporting only one smoker living in the home. For tobacco smoking households, the mean total household cigarettes smoked per day was 16.6 (SD 14.2, range 60), with the majority of smoking parents/guardians reporting smoking 20 cigarettes or less per day.

Overall, 61.9% of households did not allow smoking anywhere in or around the house, 27.8% allowed smoking outside only, and 10.3% allowed smoking inside. Many parents from non-smoking households (n = 60, 90.9%) reported that smoking was not allowed anywhere at the home, not even outside, whilst six (9.1%) parents from non-smoking households stated that smoking was allowed outside (by visiting family and friends). For smoking households, 24 (63.2%) stated that smoking was allowed outside only, and 14 (36.8%) reported that smoking was allowed inside. Out of the self-reported smoking households, 12.1% responded that smoking was allowed in the car, although seven parents failed to answer this question. Participant characteristics, split by household smoking status, are provided in the Appendix A.

#### 3.1.8. Carbon Monoxide as a Measure of Second-Hand Tobacco Smoke Exposure

Exhaled carbon monoxide (eCO) had a range of 7 ppm, with a low value of 0 ppm (below the detection limit). Mean eCO was 1.8 ppm and this was not significantly different between boys and girls (t (98) = −0.6, *p* = 0.570). Although the mean eCO was higher for children from smoking households by 17.6%, the finding was not significantly significant (t (98) = −2.3, *p* = 0.214). Exhaled CO was not correlated with the square root transformed number of cigarettes smoked per day (r = 0.157, *p* = 0.119). The concentration of eCO was highest for children from homes where smoking was permitted inside, followed by outside, then no smoking (S12), but the finding was not statistically significant (ANOVA (2,97) = 2.3, *p* = 0.104).

### 3.2. Association between Second-Hand Smoke Exposure and Children’s Cardiorespiratory Fitness

Table 2 shows the results of the unadjusted and adjusted multiple regression analyses run to predict allometrically scaled VO_2peak_ (mL·kg^−0.53^·min^−1^) from the number of cigarettes smoked per day (sqrt-cigarettes), controlling for sex, age, stature, maturation, PA, and logEIMD. Sqrt-cigarettes was not a significant predictor in the unadjusted model (R^2^ = 0.036, *F*(1,91) = 3.4, *p* = 0.068; adjusted R^2^ = 0.025). In the adjusted model, sqrt-cigarettes, sex, age, stature, and PA were significant predictors, whereas maturation and logEIMD were not. Overall, the adjusted model significantly predicted allometrically scaled VO_2peak_ (R^2^ = 0.352, *F*(7,85) = 6.6, *p* < 0.001; adjusted R^2^ = 0.299), with a moderate R^2^, explaining 29.9% of the variance. 

A follow-up linear regression analysis was undertaken in a subsample of children living in smoking households (n = 38) to explore the impact of whether indoor smoking was permitted or whether smoking was permitted outdoors only (n = 24) in the household. Table 3 shows the results of the unadjusted and adjusted multiple regression analyses run to predict allometrically scaled VO_2peak_ (mL·kg^−0.53^·min^−1^) from where smoking was permitted (indoors or outdoors only) in smoking homes, controlling for sex, age, stature, maturation, PA, and logEIMD. Smoking indoors was not a significant predictor in the unadjusted model (R^2^ = 0.011, *p* = 0.548, *F* (1,35) = 0.4; adjusted R^2^ = −0.018). In the adjusted model, only PA was a significant predictor, whereas smoking indoors, sex, age, stature, maturation, and logEIMD were not. Overall, this adjusted model did not significantly predict allometrically scaled VO_2peak_ (R^2^ = 0.297, *p* = 0.153, *F* (7,35) = 1.7; adjusted R^2^ = 0.121). 

### 3.3. Association between Second-Hand Smoke Exposure and Children’s Physical Activity 

Table 4 shows the results of the unadjusted and adjusted multiple regression analyses to predict physical activity from the number of cigarettes smoked per day (sqrt-cigarettes), controlling for sex, age, BMI, maturation, and logEIMD. Sqrt-cigarettes was not a significant predictor in the unadjusted model (R^2^ < 0.001, *F*(1,100) = 0.05, *p* = 0.826; adjusted R^2^ = −0.010). In the adjusted model, there were no statistically significant predictors of PA. Overall, the adjusted model did not predict PA (R^2^ = 0.089, *F*(5,96) = 1.9, *p* = 0.104; adjusted R^2^ = 0.042).

### 3.4. Association between Second-Hand Smoke Exposure and Markers of Respiratory Health in Children 

Several multiple regressions were run to predict FEV_1_%, FVC%, PER%, and FER from the number of cigarettes smoked per day (sqrt-cigarettes), mass, diagnosed asthma, and logEIMD. See Table 5 for the summary of the unadjusted and adjusted models for each respiratory measure (full details for each model can be found in the Appendix A).

For FEV_1_%, sqrt-cigarettes was not a significant predictor in the unadjusted model (R^2^ < 0.001, *F*(1,101) = 0.3, *p* = 0.864; adjusted R^2^ = −0.010). In the adjusted model, logEIMD was a significant predictor but sqrt-cigarettes, mass, and asthma were not. Overall, the adjusted model significantly predicted FEV1% (R^2^ = 0.138, *F*(4,100) = 3.9, *p* = 0.005; adjusted R^2^ = 0.103), although only 10.3% of the variation was explained by the model.

For FVC%, sqrt-cigarettes was not a significant predictor in the unadjusted model (R^2^ = 0.013, *F*(1,101) = 1.4, *p* = 0.247; adjusted R^2^ = 0.003). In the adjusted model, logEIMD was a significant predictor but sqrt-cigarettes, mass, and asthma were not. Overall, the adjusted model significantly predicted FVC% (R^2^ = 0.135, *F*(4,98) = 3.8, *p* = 0.006; adjusted R^2^ = 0.099), although only 9.9% of the variance was explained by the model.

For PEF%, sqrt-cigarettes was not a significant predictor in the unadjusted model (R^2^ = 0.002, *F*(1,100) = 0.2, *p* = 0.659; adjusted R^2^ = −0.008). In the adjusted model, none of the predictors were statistically significant. Overall, the adjusted model did not significantly predict PEF% (R^2^ = 0.064, *F*(4,97) = 1.7, *p* = 0.166; adjusted R^2^ = 0.025).

For FER (FEV1/FVC), sqrt-cigarettes was not a significant predictor in the unadjusted model (R^2^ = 0.015, *F*(1,99) = 1.5, *p* = 0.227; adjusted R^2^ = 0.005). In the adjusted model, none of the predictors were statistically significant. Overall, the adjusted model did not significantly predict FER (R^2^ = 0.030, *F*(4,96) = 0.7, *p* = 0.561; adjusted R^2^ = −0.010).

A multiple regression was run to predict logFeNO from the number of cigarettes smoked per day (sqrt-cigarettes), sex, age, stature, mass, diagnosed asthma, and logEIMD. Sqrt-cigarettes was not a significant predictor in the unadjusted model (R^2^ = 0.001, *F*(1,95) = 0.09, *p* = 0.760; adjusted R^2^ = −0.010). In the adjusted model, no predictors were statistically significant, and the model was not statistically significant overall (R^2^ = 0.050, *F*(7,89) = 0.7, *p* = 0.701; adjusted R^2^ = −0.025).

## 4. Discussion

Our study demonstrates that second-hand smoke exposure, as measured by the number of cigarettes smoked per day within the household, is negatively associated with children’s CRF, but no significant associations were observed for PA or respiratory measures.

### 4.1. Household Smoking 

Over one third (36.6%) of participants lived with a family member that smoked tobacco, which is significantly less than the findings of McGee et al. [33], who found 57.3% of children from the same region to have a family member that smoked. However, the current study was concerned with family members that *lived with* the participants and smoked. While smoking prevalence has also declined in the UK since 2015 [30], presently, 14.1% of adults are current smokers nationally, with smoking more common among adults in routine and manual occupations [30] with low education [79] or low SES [32,80]. The findings of the present study support the association between smoking and SES as low parental education and household deprivation were significantly associated with household smoking (see Appendix A). Although household smoking was assessed through parent/guardian self-report, the smoking prevalence findings are similar to a study by Jarvis and Feyerabend [31], who used salivary cotinine analysis and found 31.5% of children in England to be exposed to SHS. 

Exhaled carbon monoxide (eCO) was not found to be significantly different between children from smoking and non-smoking homes, although mean eCO was 17.6% higher for children from smoking homes. Exhaled CO was also not significantly correlated with the number of cigarettes smoked per day. There are several explanations for this finding, including the low sensitivity of eCO when predicting SHS exposure in children found in previous research [73], the age-related ability to perform the physically demanding test, and exposure to other environmental sources of CO prior to the test, including the road microenvironment [81], industry, and solid fuel burning [82]. Cotinine, a metabolite of nicotine, can be found in the hair, saliva, urine, and blood of individuals exposed to SHS [83,84,85] and is an alternative measure regularly used in research to determine recent active and passive smoking [86,87]. Cotinine is a sensitive and specific indicator of recent exposure to nicotine and is accepted as the best available biomarker of exposure to SHS [31]. Exhaled CO was selected as a method for establishing tobacco smoke exposure due to the low-participant burden, low cost, ease of interpretation, and instant results. However, as eCO has not shown to be an effective determinant of SHS in children, future research should seek to use salivary cotinine analysis to determine recent tobacco smoke exposure.

### 4.2. Cardiorespiratory Fitness and Physical Activity Levels 

As most of the previous research regarding CRF with children expresses fitness as mL·kg^−1^·min^−1^, it is useful to note ratio scaled VO_2peak_ in order to compare fitness profiles of our sample with those of previous relevant literature. The mean ratio scaled VO_2peak_ for boys (47.7 mL·kg^−1^·min^−1^) and girls (42.7 mL·kg^−1^·min^−1^) reported in our sample is in line with previous research with similar aged children from Northwest England. A study by Boddy et al. [66] which used a similar laboratory-based protocol to measure CRF, found means of 46.7 and 40.2 mL·kg^−1^·min for boys and girls, respectively. CRF levels in children from Northwest England have been in decline in recent years [6,7], although the similarity of the results of the present study with that of Boddy et al. [66] suggests the trend may have stabilised. Using established international thresholds of ratio scaled VO_2peak_ values [71], 86.3% of girls and 79.1% of boys in the present study reached the threshold for healthy ‘fitness’, with the remaining children below the threshold and therefore raising a ‘clinical red flag’ and at risk of cardiovascular disease. For comparison, 78% of boys and 83% of girls from 30 countries met the standards for healthy CRF [88], although CRF was estimated based on 20m shuttle run test performance. The high levels of fitness observed in our sample could be explained by high levels of participation in physical activity [3]. Indeed, the proportion of children in the present study classified as ‘active’ according to PAQ-C thresholds [89] was 95.7% for boys and 80.7% for girls. The Benítez-Porres et al. [89] thresholds suggest the use of a cut-point of 2.73 on the PAQ-C to discriminate >60 min of MVPA per day in children. Therefore, while not directly comparable due to different survey instruments, the proportion of children in the present study meeting the daily recommendation of 60 min of PA per day is far higher than the national average in England of 51% of boys and 43% of girls [13]. The mean PA score was 3.6 (SD 0.7), which is very similar to that found by Noonan et al. [90], also with children from deprived neighbourhoods of Northwest England and using the PAQ-C (3.5, SD 0.7). While the self-reported PA data should be interpreted with caution as surveys are subject to recall and desirability bias, the relatively high proportion of children classified as fit and physically active in the present study indicates that intervention efforts should focus on preventing a decline in these important health markers into adolescence and adulthood. 

### 4.3. Association of Second-Hand Smoke Exposure with Cardiorespiratory Fitness 

The present study examined the association of smoking exposure with children’s cardiorespiratory fitness using allometrically scaled VO_2peak_. As noted above, most research to date has expressed CRF as VO_2peak_ ratio scaled for mass (mL·kg^−1^·min^−1^), but expressing CRF in this way over-scales for mass and leads to spurious correlations with other health-related outcomes [70]. Recently, allometric scaling has been suggested [70] and is being used increasingly in research with youth [91,92]. Mass exponents can be generated for a sample population via log-linear regression (see methods). Mass exponents are sample-specific, and the generated exponent of the present study (0.526) is within the range of those found in previous studies [70,93,94].

Following adjustments for sex, age, stature, maturation, PA, and deprivation, the number of cigarettes smoked per day by the household was found to be negatively associated with children’s allometrically scaled VO_2peak_. A follow-up analysis was conducted to explore the effect of smoking indoors compared to outdoors only on children from smoking households’ CRF. Although smoking indoors was not found to be a statistically significant predictor within the model, the negative B coefficient indicates a negative relationship between smoking indoors and children’s CRF. As the number of parents that reported smoking indoors was low (n = 14), further research with a larger sample size is warranted given that smoking indoors in the household is likely to be more harmful to the child than smoking being permitted outdoors only. Although no prior research has yet examined the impact of SHS exposure on children’s laboratory measured VO_2peak_, the results are in line with Magnússon et al. [38], who have found children’s CRF (measured by maximal cycle ergometer test) to be significantly reduced for those with smoking parents. Kaymaz et al. [40] have also shown that children exposed to parental smoking have reduced performance on the six-minute walk test. The mechanism by which SHS exposure reduces VO_2peak_ cannot be determined from the present study, but key components of tobacco smoke such as CO and particulate matter have each been shown to individually impact CRF. CO, which has a higher affinity for haemoglobin than oxygen, acutely decreases aerobic capacity through hypoxaemia of peripheral tissues due to haemoglobin-bound CO [37]. Particulate matter exposure causes systemic inflammation and increased oxidative stress, leading to impaired cardiovascular, immune, and pulmonary function [95] and reduced exercise performance [96]. A vast body of research relating to the detrimental effects of tobacco smoke exposure has emerged since the pioneering work of Doll and Hill in the 1950s [97,98], but recently, research in the emerging field of epigenetics has observed the transgenerational effects of tobacco smoke, whereby paternal and maternal smoking results in changes in DNA methylation for the offspring that persist many years after exposure [99,100]. 

It is possible that there is a dose–response relationship between SHS exposure and CRF, as dose–response relationships have been observed between SHS exposure and a number of other health-related variables including birthweight, sudden infant death syndrome, cognitive and behavioural problems, respiratory issues, childhood obesity, and increased blood pressure 18 years post-exposure [101,102,103]. In the present study, the range for the number of cigarettes smoked per household per day was large at 65, and the data were positively skewed as the majority of participants were from non-smoking households. Most of the participating smoking parents/guardians smoked 20 or less cigarettes per day, with very few households smoking more than 20. Future work should aim to include more children from heavily smoking households, either through a larger sample size or targeted recruitment. Additionally, as discussed above, cotinine testing has a very high sensitivity and specificity for SHS exposure and might enable better quantification of a potential dose–response relationship between SHS and CRF in children.

Children may be more susceptible to the effects of SHS due to their increased respiratory rates and immature and developing organs [24]. Furthermore, children may be especially vulnerable if exposed prenatally [103,104]. Although in utero exposure was not within the scope of this study, there is a need for longitudinal studies to examine the impact of SHS exposure across the life-course. It would be of value to understand whether the apparent detrimental effects of SHS seen in children’s fitness in the present study track into late adolescence and adulthood, and whether the effects persist even after SHS exposure has ceased. 

### 4.4. Association of Second-Hand Smoke Exposure with Physical Activity

Despite the negative association between SHS exposure and CRF, we found no relationship between the number of cigarettes smoked in the home and children’s self-reported PA. These findings are encouraging as results indicate SHS exposure is not impacting children of smoking households’ engagement in PA, which has beneficial implications for subsequent disease risk. Nevertheless, recent qualitative research indicates that household smoking status may influence children’s perceptions, barriers, facilitators, and beliefs surrounding PA and exercise [43]. Children from smoking households rated metabolically demanding activities such as running as more difficult than children from non-smoking homes, and indicated a preference for less strenuous activities [43]. The same study also showed that children from non-smoking households demonstrated greater awareness of PA guidelines, referred to extrinsic motivators of PA and the health benefits of fitness, and had considerations for the future self in terms of PA and fitness. As per the Youth Physical Activity Promotion Model [105], personal demographics (age, gender, ethnicity, SES), enabling factors (fitness, skills, access, environment), reinforcing factors (family, peer, and couch influence), and predisposing factors (perceived competence, self-efficacy, enjoyment, beliefs, attitudes) can all influence PA in children. PA is an established determinant of CRF [74], and an active child is more likely to be fit and healthy [106,107]. However, as the reverse causation hypothesis for obesity and PA implies a positive feedback loop [108], low CRF may similarly discourage participation in PA, further decreasing CRF in the inactive child.

### 4.5. Markers of Respiratory Health 

#### 4.5.1. Association between Second-Hand Smoke Exposure and Spirometry Outcomes 

Across the sample, and for all measures (FEV1%, FVC%, PEF%, FER), spirometry values were below the predicted values based on children’s age, sex, stature, and ethnicity by 10.3–24.7%, suggesting lower than average lung function across the whole sample. Participant cooperation, effort, experience, and practice are important factors when undertaking spirometry testing, and children require encouragement and practice to successfully undertake the forced manoeuvres required for a valid test [109]. Despite the high levels of encouragement from the trained research team, some children may not have cooperated fully and therefore achieved sub-optimal spirometry performance. 

Linear regression analysis showed that the number of cigarettes smoked per day was not a significant predictor of FEV_1_%, FVC%, PEF%, or FER, following adjustments for mass, asthma, and deprivation. For the FEV_1_% and FVC% models, deprivation (EIMD) was the only statistically significant predictor, indicating higher deprivation is associated with decreased lung function. All participants’ postcodes were within the lowest four EIMD deciles (high and medium deprivation), and 85.5% of participant postcodes were within the lowest two deciles. Socioeconomic status is an established determinant of lung function [78,110], and the low spirometry values across the sample may be reflecting the low SES of the sample. Indoor and outdoor air quality are associated with respiratory health [111], and living near major roadways is associated with decreased FVC and increased FeNO [112]. Poverty and environmental exposures may explain the ethnic differences for spirometry performance observed within the literature [113]. In the present study, spirometry values were normalised by a factor of 0.9 for black children and 0.95 for children of other ethnicities as per Korotzer et al. [62]. 

Our results contrast with research that has demonstrated a negative association between SHS exposure and lung function. A large amount of research has shown SHS exposure to be detrimental to lung function [114,115,116,117,118], with the effects of early life exposure observed decades later [25]. Li et al. [117] suggest that results demonstrating the impact of SHS on children’s lung health should be interpreted considering in utero exposure. Data on in utero exposure were not collected in the present study, but previous research indicates that in utero exposure to tobacco smoke is especially detrimental to lung function (reduced expiratory flow, mid-expiratory flow, forced vital capacity, fractional exchange ratio), likely due to the effects of SHS on development and growth [118,119]. Future work could incorporate data on in utero exposures, as well as monitoring lung function during and after maximal exercise, and may reveal exercise-related variations in lung function for children exposed to tobacco smoke both in utero and ex utero. 

#### 4.5.2. Association between Second-Hand Smoke Exposure and Fractional Exhaled Nitric Oxide

The range of FeNO concentrations in our sample was large (147 ppb), and most children (70.6%) had low levels (<20 ppb) of FeNO, 13.7% had intermediate levels (20–35 ppb), and 15.7% had high levels (>35 ppb). High levels of FeNO indicate eosinophilic airway inflammation, which itself may indicate asthma [58]. Although the FeNO concentrations of diagnosed asthmatics (n = 9) were slightly elevated compared to non-diagnosed asthmatics, this was not statistically significant. However, this may indicate that asthma was successfully being treated in those diagnosed [120]. FeNO concentrations can also be influenced by recent ingestion of food and drink, foods high in nitrates [121], rhinovirus infection, allergic rhinitis [122], and genetics [123].

Nitric oxide (NO) is important for metabolic regulation during exercise as a modulator of blood flow, regulating muscle contraction and influencing muscle glucose uptake [124]. Nitric oxide bioavailability is associated with increased exercise performance in untrained individuals, including a reduced O_2_ cost of low-intensity exercise and improved exercise time to exhaustion [125,126], with exercise training elevating NO bioavailability [124]. SHS exposure may reduce CRF through the action of particulate matter on the bioavailability of NO, an important and potent vasodilator. However, after adjusting for sex, age, mass, stature, asthma, and deprivation, no association was found between smoking exposure and FeNO. None of the predictors in the adjusted model, including the number of cigarettes, sex, age, mass, stature, asthma, or EIMD, were significant in the model for FeNO. For active and passive smokers, the association between FeNO and airway inflammation is complicated. Although FeNO is increased in untreated adult asthmatics who smoke, FeNO concentrations are generally reduced in smokers and individuals exposed to SHS [127,128]. Increasing cotinine levels have been found to be associated with a progressive reduction in FeNO and an increase in blood eosinophil count in healthy individuals aged 6–80 years [129]. The mechanism by which tobacco smoke reduces FeNO is likely to be through the reduction in the enzymatic activity of nitric oxide synthase, in combination with superoxides (found in tobacco smoke in high concentrations), which react with NO to produce active nitrogen species [130]. Therefore, FeNO is reduced in active and passive smokers due to the suppression of production and elimination of NO. Future work should look to understand how FeNO changes in exercising children exposed to SHS. As NO is important for several biological and exercise-related pathways, the reduction of NO in SHS-exposed individuals may be significant in relation to CRF and warrants further research in paediatric populations.

### 4.6. Strengths and Limitations

This study is the first to examine the association between SHS exposure and children’s CRF, PA, and respiratory measures including FeNO and spirometry. CRF was determined by direct measurement of VO_2peak_ through a laboratory-based treadmill protocol, which is the ‘gold standard’ measure of young people’s CRF [70]. The sample population of this study are representative of 9–11-year-old children from deprived areas in Merseyside, UK, providing valuable information relating to SHS exposure and health markers for this demographic. The study nonetheless includes some limitations. Due to the high research saturation of local primary schools and the contentious nature of the project, the study achieved a relatively small sample size of 104 participants (3% participation rate), including 38 children from smoking households. Furthermore, the relatively high proportion of children classified as fit in the present study may also be due to bias, where predominantly active children with confidence in their abilities volunteer to participate in the study. While the sample population of this study are representative children from deprived areas of the UK, future research should aim to include children from a greater variety of SES backgrounds, age groups, and geographic locations. Household smoking status and child PA were measured via surveys, which are subject to recall errors and desirability bias and may have led to under/over estimation of these behaviours. Device-based measurement of PA could have provided further information regarding PA behaviour, a key determinant of CRF. Furthermore, while the use of eCO as a method of SHS exposure quantification was selected due to the low-participant burden, low cost, ease of interpretation, and instant results, future studies should seek to use cotinine testing to better determine SHS exposure. 

## 5. Conclusions

To the authors’ knowledge, this is the first study to examine the association between SHS exposure and children’s CRF, PA, and respiratory health. An important finding was that the number of cigarettes smoked per day in each household was a significant and negative predictor of allometrically scaled VO_2peak_ (mL·kg^−0.53^·min^−1^). No associations were found between SHS exposure and physical activity or between SHS exposure and spirometry and FeNO respiratory health measures. CRF is a global measure of health, and these findings are indicative of lower health status in children from smoking households. Low CRF is associated with a plethora of negative health outcomes and as fitness tracks into adulthood, efforts should be made to improve CRF during childhood. Reducing SHS exposure may be an effective measure for improving CRF in children from smoking households, and a potential avenue for intervention aiming to improve CRF in low SES populations. Future work should aim to incorporate cotinine testing and the use of a larger sample of children exposed to SHS to determine and quantify the potential dose–response relationship between SHS and CRF. Additionally, research is required to determine the mechanism by which SHS exposure is detrimental to children’s CRF, and the use of longitudinal research is required to uncover long-term impacts of SHS exposure and children’s CRF.

## Figures and Tables

**Table 1 ijerph-18-11445-t001:** Descriptive statistics for the sample.

	N	Minimum	Maximum	Mean	SD
*Anthropometry*					
Decimal age (years)	104	8.5	11.5	10.1	0.6
Maturation (years to PHV)	103	−4.1	0.1	−2.2	1.0
Stature (cm)	103	122.0	158.0	141.7	6.6
Mass (kg)	103	22.8	66.0	38.2	9.2
BMI (kg·m^−2^)	103	13.2	30.5	19.0	3.9
*Cardiorespiratory fitness*					
VO_2peak_ (mL·min^−1^)	94	843.0	2399.0	1659.5	307.9
VO_2peak_ (mL·kg^−1^·min^−1^)	94	24.8	59.5	45.0	7.7
VO_2peak_ (mL·kg^−0.53^·min^−1^)	94	157.5	322.6	247.2	36.3
*Respiratory health*					
FEV_1_ (%)	103	43.9	131.7	83.0	17.2
FVC (%)	103	44.3	136.0	89.0	19.7
PEF (%)	103	33.6	155.7	75.3	21.0
FEV_1_/FVC	103	52.7	100.0	89.6	11.0
FeNO * (ppb)	102	<5	147	15.9	33.4
*Physical activity*					
PAQ-C	103	2.1	5.0	3.6	0.7
*SHS exposure*					
eCO (ppm)	100	0	7	1.8	1.2
Cigarettes per day	104	0	65	5.5	10.8
*Deprivation*					
EIMD rank *	104	69	25,530	1427	5652

PHV = peak height velocity, BMI = body mass index; Spirometry values expressed as percentage of predicted values for sex, age, ethnicity, and height. FEV_1_ = forced expiratory volume in 1 s, FVC = forced vital capacity, PEF = peak expiratory flow, FeNO = fractional exhaled nitric oxide, eCO = exhaled carbon monoxide, EIMD = English Indices of Multiple Deprivation, maturation = years from peak height velocity. Physical activity (PAQ-C) is scored between 1 and 5, with 5 being the most active, SHS = Second-hand smoke exposure, EIMD = English Indices of Multiple Deprivation. * Indicates geometric mean and geometric standard deviation.

**Table 2 ijerph-18-11445-t002:** Linear regression models examining association between the number of cigarettes smoked per day per household and allometrically scaled VO_2peak_ (mL·kg^−0.53^·min^−1^).

Model and Predictor	Unstandardised Coefficient (B)	95% Confidence Interval		Standard Error of B	Significance
		Lower Bound	Upper Bound		
*Unadjusted model* (R^2^ = 0.036, *p* = 0.068, F = 3.4)					
Constant	252.3	243.1	261.5	4.6	**<0.001**
Sqrt-cigarettes	−3.7	−7.6	0.3	2.0	0.068
*Adjusted model* (R^2^ = 0.352, *p* < 0.001, F = 6.6)					
Constant	−113.7	−331.9	104.5	109.7	0.303
Sqrt-cigarettes	−3.8	−7.3	−0.4	1.7	**0.030**
Sex	−26.2	−49.0	−3.4	11.5	**0.025**
Age (years)	12.2	0.4	23.9	5.9	**0.042**
Stature (cm)	1.3	0.1	2.5	0.6	**0.036**
Maturation (years to PHV)	0.1	−12.4	12.4	6.2	0.998
Physical activity	15.4	6.0	24.8	4.7	**0.002**
LogEIMD	5.2	−5.7	16.0	5.5	0.347

Abbreviations: Sqrt-cigarettes = the square root of the total number of cigarettes smoked per household per day; PHV = peak height velocity; logEIMD = log-transformed English Indices of Multiple Deprivation (based on household postcode).

**Table 3 ijerph-18-11445-t003:** Linear regression models examining association between indoor and outdoor smoking and allometrically scaled VO_2peak_ (mL·kg^−0.53^·min^−1^) among children from smoking households (n = 38).

Model and Predictor	Unstandardised Coefficient (B)	95% Confidence Interval		Standard Error of B	Significance
		Lower Bound	Upper Bound		
*Unadjusted model * (R^2^ = 0.011, *p* = 0.548, F = 0.4)					
Constant	241.3	226.4	256.3	7.4	**<0.001**
Smoke indoors	−7.4	−32.3	17.5	12.2	0.548
*Adjusted model* (R^2^ = 0.297, *p* = 0.153, F = 1.7)					
Constant	136.3	−318.4	591.0	222.0	0.544
Smoking indoors	−17.0	−41.2	7.1	11.8	0.159
Sex	−26.3	−62.0	9.3	17.4	0.142
Age (years)	−0.7	−24.2	22.8	11.5	0.951
Stature (cm)	0.6	−1.9	3.2	1.2	0.611
Maturation (years to PHV)	5.6	−14.4	25.7	9.8	0.570
Physical activity	20.4	4.7	36.2	7.7	**0.013**
LogEIMD	−7.5	−32.0	17.0	12.0	0.538

Abbreviations: PHV = peak height velocity; logEIMD = log-transformed English Indices of Multiple Deprivation (based on household postcode). Bold for statistically significant findings (*p* < 0.05).

**Table 4 ijerph-18-11445-t004:** Linear regression models examining association between smoking exposure and self-reported physical activity.

Model and Predictor	Unstandardised Coefficient (B)	95% CI		Standard Error of B	Significance
		Lower Bound	Upper Bound		
*Unadjusted model * (R^2^ < 0.001, *p* = 0.826, *F* = 0.05)					
Constant	3.64	3.47	3.81	0.08	**<0.001**
Sqrt-cigarettes	0.01	−0.07	0.08	0.04	0.826
*Adjusted model * (R^2^ = 0.089, *p* = 0.104, *F* = 1.9)					
Constant	6.54	3.05	10.03	1.76	**<0.001**
Sqrt-cigarettes	0.03	−0.05	0.11	0.04	0.456
Sex	−0.29	−0.80	0.21	0.25	0.255
Age (years)	−0.21	−0.47	0.04	0.13	0.100
BMI (kg·m^−2^)	−0.03	−0.07	0.01	0.02	0.104
Maturation (years to PHV)	0.05	−0.22	0.31	0.13	0.732
LogEIMD	0.04	−0.20	0.28	0.12	0.751

Abbreviations: Sqrt-cigarettes = the square root of the total number of cigarettes smoked per household per day; BMI = body mass index; PHV = peak height velocity; logEIMD = log-transformed English Indices of Multiple Deprivation (based on household postcode). Bold for statistically significant findings (*p* < 0.05).

**Table 5 ijerph-18-11445-t005:** Summary of spirometry and FeNO linear regression models.

Model	Model R^2^	Model Significance, *p*	Unstandardised Coefficient (B) for Sqrt-Cigarettes	95% Confidence Interval		Standard Error of B	Significance of B, *p*
				Lower Bound	Upper Bound		
FEV_1_% unadjusted	<0.001	0.864	−0.2	−2.0	1.7	0.9	0.864
FEV_1_% adjusted	0.138	0.005	0.4	−1.4	2.2	0.9	0.660
FVC% unadjusted	0.013	0.247	−1.2	−3.3	0.9	1.1	0.247
FVC% adjusted	0.135	0.006	−0.7	−2.8	1.3	1.0	0.494
PEF% unadjusted	0.002	0.659	−0.6	−2.7	1.6	1.1	0.608
PEF% adjusted	0.064	0.166	0.0	−2.2	2.1	1.1	0.965
FER unadjusted	0.015	0.227	0.6	−0.4	1.7	0.5	0.227
FER adjusted	0.030	0.561	0.7	−0.4	1.8	0.5	0.204
FeNO unadjusted	0.001	0.760	−0.01	−0.09	0.07	0.04	0.760
FeNO adjusted	0.050	0.701	−0.01	−0.10	0.08	0.04	0.818

Abbreviations: Sqrt-cigarettes = the square root of the total number of cigarettes smoked per household per day; FEV_1_ = forced expiratory volume in 1 s; FVC = forced vital capacity; PEF = peak expiratory flow; FER = forced expiratory ratio (FEV_1_/FVC); FeNO = ln fractional exhaled nitric oxide. FEV_1_, FVC, and PEF expressed as percentages of predicted for sex, age, height, and ethnicity. FEV_1_%, FVC%, PEF%, and FER models adjusted for mass, asthma diagnosis, and EIMD. FeNO models adjusted for sex, decimal age, mass, stature, asthma diagnosis, and EIMD.

## Data Availability

The data in this study are available upon reasonable request from the corresponding author. The data are not publicly available due to privacy and ethical considerations.

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
