# Peer review of "Associations between Second-Hand Tobacco Smoke Exposure and Cardiorespiratory Fitness, Physical Activity, and Respiratory Health in Children"

_ijerph, 2021, doi:10.3390/ijerph182111445_

Round 1

Reviewer 1 Report

The authors have studied the associations between ETS with cardiorespiratory fitness (CRF) and physical activity (PA) in 104 children with only 38 children from smoking households. Results showed that SHS exposure was negatively associated with allometrically scaled V̇O2peak.

The main criticism is on the definition of ETS exposure, that initially was based on exhaled carbon monoxide (eCO) and then on the number of cigarettes smoked per day from parents.  Unfortunately, the estimation of these variables was not performed accurately, Authors have missed  to correct for how long time the children were really exposed to tobacco smoke at home. Models studying allometrically scaled V̇O2peak in indoor or outdoor were not reported for the 38 smoking households. 

The measurements of urinary cotonine or the use of other markers of ETS exposure are necessary to confirm their results and exposure status of children. 

In the discussion, the suggested role of ETS exposure on weight gain of chidren is not supported from study results.

Author Response

General response to reviewers

We would like to thank the reviewers for their thoughts and suggestions within this first round of review. Within this revision, changes have been made as requested throughout the manuscript. We hope that these additions and amendments address the reviewers’ thoughts and suggestions. We provide a detailed point by point response below and have highlighted the amendments within the submitted manuscript in yellow.

Reviewer 2 Report

This paper is written well.

However, introduction and discussion are too long to find what is important.

Authors should clarify the issues.

I think this paper is good for accept after minor revision.

Author Response

(The authors gave the same response as above.)

Reviewer 3 Report

Since all the children in the study are from deprived areas, perhaps adding this information to the title and/or abstract should be considered.

Throughout the text there are inconsistencies when referring to electronic cigarettes. The terms electronic cigarettes or e-cig and are preferable to "vaper". Lines 152-155 should be rewritten according to this nomenclature.

If, after performing the logarithmic transformation of a variable, normality can be assumed, the geometric mean and geometric standard deviation should be reported. This applies to both Table 1 and the results.

Author Response

(The authors gave the same response as above.)

Round 2

Reviewer 1 Report

The manuscript has been appropriately revised and modified by the authors according to the requests of the reviewers and warrant publication